# Machine learning and LHC event generation

Anja Butter[1,2], Tilman Plehn[1], Steffen Schumann[3], Simon Badger[4], Sascha Caron[5,6]
Kyle Cranmer[7,8], Francesco Armando Di Bello[9], Etienne Dreyer[10], Stefano Forte[11],
Sanmay Ganguly[12], Dorival Gonçalves[13], Eilam Gross[10], Theo Heimel[1],
Gudrun Heinrich[14], Lukas Heinrich[15], Alexander Held[16], Stefan Höche[17],
Jessica N. Howard[18], Philip Ilten[19], Joshua Isaacson[17], Timo Janßen[3], Stephen Jones[20],
Marumi Kado[9,21], Michael Kagan[22], Gregor Kasieczka[23], Felix Kling[24], Sabine Kraml[25],
Claudius Krause[26], Frank Krauss[20], Kevin Kröninger[27], Rahool Kumar Barman[13],
Michel Luchmann[1], Vitaly Magerya[14], Daniel Maitre[20], Bogdan Malaescu[2],
Fabio Maltoni[28,29], Till Martini[30], Olivier Mattelaer[28], Benjamin Nachman[31,32],
Sebastian Pitz[1], Juan Rojo[6,33], Matthew Schwartz[34], David Shih[25], Frank Siegert[35],
Roy Stegeman[11], Bob Stienen[5], Jesse Thaler[36], Rob Verheyen[37],
Daniel Whiteson[18], Ramon Winterhalder[28], and Jure Zupan[19]

## Abstract

First-principle simulations are at the heart of the high-energy physics research program. They link the vast data output of multi-purpose detectors with fundamental theory predictions and interpretation. This review illustrates a wide range of applications of modern machine learning to event generation and simulation-based inference, including conceptional developments driven by the specific requirements of particle physics. New ideas and tools developed at the interface of particle physics and machine learning will improve the speed and precision of forward simulations, handle the complexity of collision data, and enhance inference as an inverse simulation problem.

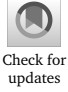

**1** Institut für Theoretische Physik, Universität Heidelberg, Germany
**2** LPNHE, Sorbonne Université, Université Paris Cité, CNRS/IN2P3, Paris, France
**3** Institut für Theoretische Physik, Georg-August-Universität Göttingen, Germany
**4** Physics Department, Torino University and INFN Torino, Italy
**5** IMAPP, Radboud Universiteit Nijmegen, The Netherlands
**6** Nikhef Theory Group, Nikhef, Amsterdam, The Netherlands
**7** Center for Cosmology and Particle Physics, New York University, New York, NY USA
**8** Center for Data Science, New York University, New York, NY USA
**9** Universita di Roma Sapienza, Roma, INFN, Italy
**10** Weizmann Institute of Science, Rehovot, Israel
**11** Dipartimento di Fisica, Università di Milano and INFN Sezione di Milano, Italy
**12** ICEPP, University of Tokyo, Japan
**13** Department of Physics, Oklahoma State University, Stillwater, OK, USA
**14** Institut für Theoretische Physik, Karlsruher Institut für Technologie, Germany
**15** Physik-Department, Technische Universität München, Germany
**16** Department of Physics, New York University, New York, NY USA
**17** Theoretical Physics Division, Fermi National Accelerator Laboratory, Batavia, IL, USA


18 Department of Physics & Astronomy, UC Irvine, Irvine, CA, USA
19 Department of Physics, University of Cincinnati, Cincinnati, OH, USA
20 IPPP, Physics Department, Durham University, Durham, UK
21 Université Paris-Saclay, CNRS/IN2P3, IJCLab, Orsay, France
22 Fundamental Physics Department, SLAC National Accelerator Laboratory, USA
23 Institut für Experimentalphysik, Universität Hamburg, Germany
24 Deutsches Elektronen-Synchrotron DESY, Germany
25 Univ. Grenoble Alpes, CNRS, Grenoble INP, LPSC-IN2P3, France
26 NHETC, Dept. of Physics and Astronomy, Rutgers University, Piscataway, NJ, USA
27 Department of Physics, TU Dortmund University, Germany
28 CP3, Université Catholique de Louvain, Louvain-la-Neuve, Belgium
29 Dipartimento di Fisica e Astronomia, Università di Bologna, Italy
30 Institut für Physik, Humboldt-Universität zu Berlin, Germany
31 Physics Division, Lawrence Berkeley National Laboratory, Berkeley, CA, USA
32 Berkeley Institute for Data Science, University of California, Berkeley, CA, USA
33 Department of Physics and Astronomy, Vrije Universiteit Amsterdam, The Netherlands
34 Department of Physics, Harvard University, Cambridge MA, USA
35 Institute of Nuclear and Particle Physics, Technische Universität Dresden, Germany
36 Center for Theoretical Physics, MIT, Cambridge, MA, USA
37 Department of Physics & Astronomy, University College London, UK

## Contents

# 1 Introduction

The defining goal of particle physics is to understand the fundamental nature of elementary particles and their interactions. The outcome of a particle physics measurement is expressed in terms of a quantum field theory Lagrangian and its parameters. The great experimental strength of collider-based particle physics is the availability of a huge amount of data and measurements in combination with a well-controlled environment. The theoretical and experimental poles are linked through precision simulations, starting from the Standard Model or a hypothetical Lagrangian, generating particle-level events, and eventually simulating the detector. The simulation chain realized by the standard LHC event generators [1–5] and illustrated in Fig. 1, should be based on first-principles physics rather than empiric modeling. For these simulations precision and speed are essentially two sides of the same medal. A detailed discussion of these traditional methods can be found in a parallel review, Ref. [6]. Adding modern machine learning to the numerics toolbox has the potential to provide the simulations needed for the LHC Run 3 and HL-LHC [7], as well as future energy frontier machines.

From a fundamental physics perspective there exist three distinctly different kinds of measurements at the LHC. First, basic and purely experimental measurements should be as independent of theory considerations and first-principle simulations as possible, to avoid expiration dates. Their problem is that they provide no information about fundamental physics. These basic measurements benefit from modern machine learning for instance in understanding the data and calibrating the detectors. A second class of measurements is supplemented with a fundamental theory interpretation framework. Examples are well-defined inclusive production rates, like fiducial or total cross sections. They can be compared to predictions from perturbative quantum field theory. When we expect to find agreement with the Standard Model, modern machine learning can help us in using these measurements to extract parton densities or improve our Monte Carlo simulations. A third kind of measurement reflects our goal to further our understanding of fundamental physics by comparing data to predictions from perturbative or non-perturbative quantum field theory. We assume that interesting physics signals hide in specific kinematic regions. Here, we can search for deviations between the Standard Model predictions and experimental results, measure Standard Model parameters or higher-dimensional Wilson coefficients, and aim for anomalies and eventually a proper discovery. Such measurements of all possible features in the vast phase space of LHC collisions require precision simulations, specifically theory-based event generators. We will show how all of these aspects benefit significantly from the application of modern machine learning methods.

The challenges for event generators are, first of all, defined by the increase of the LHC luminosity and the expected advances in experimental precision and reach. Going from the Run 2 dataset of 139 fb$^{-1}$ to the projected HL-LHC dataset of 4 ab$^{-1}$ suggests that experimental uncertainties at and below the percent level will become standard and need to be matched by theory predictions, to allow for any kind of precision measurement. The same increase in rate will allow us to probe more and more exotic kinematic regions, with the hope of finding

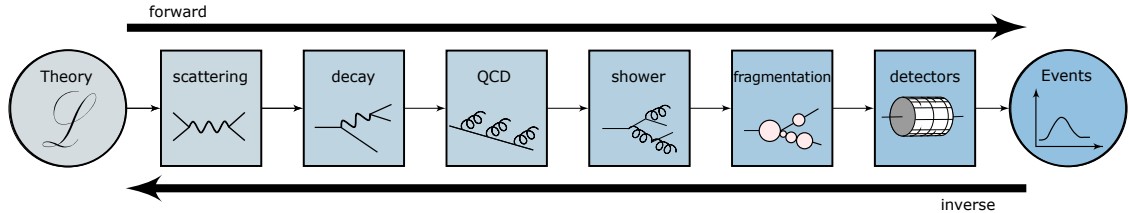

Figure 1: Illustration of the LHC simulation chain. The forward direction is discussed in Secs. 2 and 3, while the inverse simulation is the topic of Sec. 4.

hints for new particles and interactions. The higher rates and an improved experimental understanding will also allow us to study more and more complex signatures with an increasing multiplicity of backgrounds and potential signals. Each of these aspects poses a challenge to the established event generators, and we will discuss ways modern machine learning can help us meet them in Sec. 2. Next, we will introduce end-to-end (soup-to-nuts) ML-generators, similar in structure to ML-detector simulations in Sec. 3. Finally, we discuss conceptual benefits from modern machine learning, for instance related to an invertible simulation chain and simulation based inference, in Sec. 4. Because the main purpose of this report is to show new, ML-driven developments in event generation, we refer to the main event generator Snowmass white paper for a list of references and a detailed discussion of the physics background and the classical approaches.

## 2 Machine Learning in event generators

Current multi-purpose event generators feature a modular structure, that reflects the factorization property of physics aspects at very different relevant energy scales [1–5]. While the highest energy transfers, i.e. the hard process and QCD parton showers, can be treated by perturbative methods, phenomenological models are used to account for the hadronization transition, as well as non-trivial secondary interactions. The increase in perturbative precision needed to address the physics challenges posed by current and future collider experiments, adds a sizeable number of more specialized numerical codes to the simulation toolbox. This includes, for example, dedicated codes to construct and evaluate higher-order tree-level or loop amplitudes. Modern machine learning techniques can improve all aspects of event generation, ultimately making it more resource efficient and opening paths to yet more versatile and accurate predictions. This includes important ingredients to precision predictions such as parton densities and fragmentation functions, where neural network (NN) techniques are routinely used already. First steps towards modeling the hadronization process with ML techniques have been presented in [8]. For the tuning of non-perturbative simulation parameters, including an underlying event model, NN-based approaches have recently shown promise [9].

### 2.1 Phase space sampling

The core of any scattering event simulation is the assumed hard process configuration or partonic scattering event. These are described by QFT transition amplitudes, where the physics demands of the LHC experiments require us to consider high-multiplicity final states and one- or even two-loop QCD and/or EW corrections. The complexity of the resulting matrix elements and the dimensionality of their phase space severely challenge the integration of cross sections

Table 1: Results for sampling the top decay width, the total cross section of top-pair production and decay in $e^+e^-$ collisions, and $gg \to 3g$ and $4g$ production. Shown are the integral estimate, $E_N$, and the unweighting efficiency, $\epsilon_{\mathrm{uw}}$, for a standard importance sampler (Uniform), VEGAS, and NN-based optimization [10].

|  | top decays | | top-pair production | |  $gg \to 3g$ | |  $gg \to 4g$ | |
|---|---|---|---|---|---|---|---|---|
| Sample | $\epsilon_{\mathrm{uw}}$ | $E_N$ [GeV] | $\epsilon_{\mathrm{uw}}$ | $E_N$ [fb] | $\epsilon_{\mathrm{uw}}$ | $E_N$ [fb] | $\epsilon_{\mathrm{uw}}$ | $E_N$ [fb] |
| Uniform | 59 % | 0.1679(2) | 35 % | 1.5254(8) | 3.0 % | 24806(55) | 2.7 % | 9869(20) |
| VEGAS | 50 % | 0.16782(4) | 40 % | 1.5251(1) | 27.7 % | 24813(23) | 31.8 % | 9868(10) |
| NN | 84 % | 0.167865(5) | 78 % | 1.52531(2) | 64.3 % | 24847(21) | 48.9 % | 9859(10) |

and the generation of partonic momentum configurations. Modern NN-techniques are ideally suited to assist in these tasks. The standard technique used so far is based on importance sampling, employing mappings $\vec{y} : V \to U \subseteq \mathbb{R}^d$ for phase space integrals

$$I = \int_V d^d x\, f(x) = \int_U d^d y \left. \frac{f(x)}{g(x)} \right|_{x \equiv x(y)}, \qquad \text{with} \quad \left| \frac{\partial y(x)}{\partial x} \right| = g(x). \tag{1}$$

It can be chosen such that $f/g \approx$ const, to reduce the variance of the Monte Carlo integral estimate. However, for complex matrix elements and high-dimensional phase spaces it is often not possible to find a single function $g$ that approximates the target function $f$ sufficiently well. Therefore, event generators use a multi-channel approach with independent mappings $\vec{y}_i$ for each channel $i$. Defining a total density $g(x) = \sum_i \beta_i\, g_i(x)$, with $\sum_i \beta_i = 1$ and $0 \leq \beta_i \leq 1$, where $\beta_i$ are the channel weights, the phase space integral can be parametrized as

$$I = \int_V d^d x\, f(x) = \sum_i \int_V d^d x\, \beta_i\, g_i(x) \frac{f(x)}{g(x)} = \sum_i \int_{U_i} d^d y_i\, \beta_i \left. \frac{f(x)}{g(x)} \right|_{x \equiv x(y_i)}. \tag{2}$$

Two ML-based approaches to phase space integration and event generation can be distinguished. The first directly hooks into existing phase space integrators and uses trainable maps given for example by bijective normalizing flows to redistribute input random variables to the mapping functions $\vec{y}_i$ and better adapt to the integrand [10–16]. After an initial adaptation phase these integrators can efficiently be used for generating weighted or unweighted events. However, the very expressive NN-transformations can also deal with non-factorizable phase space structures and correlations. Promising results in terms of efficiency improvements and speed gains have been reported, see for example Tabs. 1 and 2. However, in particular for high-multiplicity processes with non-trivial topologies the effective gains when comparing to the established methods can fall below unity, cf. Tab 2. Therefore, next steps will be to better combine NN-based approaches with multi-channel integrators [10, 14]. For example, one can allow the channel weights to be phase space dependent, $\beta_i \to \alpha_i(x)$, and solely start from the condition $\sum_i \alpha_i(x) = 1$ and $0 \leq \alpha_i(x) \leq 1$,

$$I = \int_V d^d x\, f(x) = \sum_i \int_V d^d x\, \alpha_i(x) f(x) = \sum_i \int_{U_i} d^d y_i\, \alpha_i(x) \left. \frac{f(x)}{g_i(x)} \right|_{x \equiv x(y_i)}. \tag{3}$$

In fact, Eqs. (2) and (3) are mathematically equivalent, connected by $\alpha_i(x) = \beta_i\, g_i(x)/g(x)$. In NN-optimized event generation, Eq.(3) splits the optimization task into learning appropriate phase space mappings for each channel and training another network to find optimal weights $\alpha_i(x)$ to connect all channels. This separation has two advantages: (i) possible missing correlations between the different channels can be described and recovered by the phase-space dependent channel weights, and (ii) the second network allows for a more flexible parametrization as it does not need to be bijective.

A second approach to ML-assisted phase space sampling is based on directly learning the phase space distribution of events from input training samples, either weighted or unweighted. Solutions employ autoregressive flows [17], generative adversarial networks (GANs) [18–21], or variational autoencoders (VAEs) [20]. This motivates R&D to improve training through differentiable programming; by merging matrix element codes with automatic differentiation [22], i.e. the automatic generation of derivatives of programs that is the backbone of neural networks software frameworks. The gradients of matrix elements can be evaluated and used as additional information for training generative models. Initial studies using differentiable matrix elements from MADJAX have explored extending normalizing flow training with schemes uniquely enabled by the ability to automatically compute matrix element gradients [23], and show promise in terms of improving modeling and reducing the needed scale of simulated datasets for training.

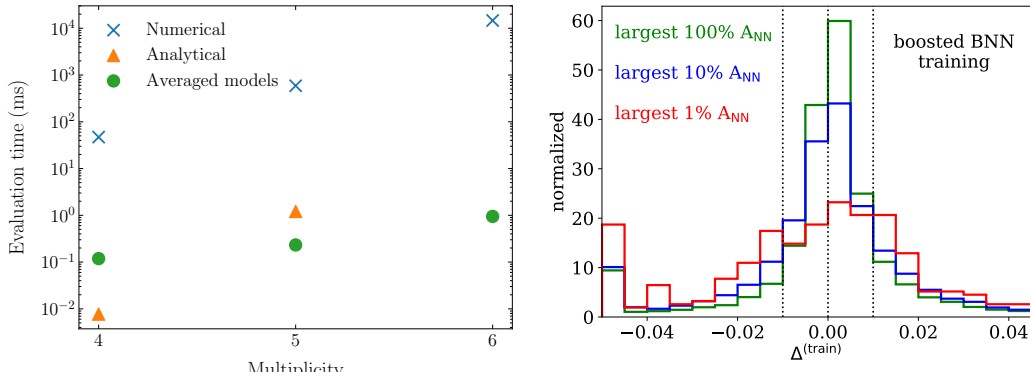

Figure 2: Left: comparison of the evaluation times for loop-induced amplitudes for $gg \to \gamma\gamma+$ jets. NN-interpolation times include the averaging over ensembles for the uncertainty estimate. Figure from Ref. [29]. Right: precision $\Delta^{\text{train}} = |\mathcal{M}|^2_{\text{NN}}/|\mathcal{M}|^2_{\text{train}} - 1$ for the process $gg \to \gamma\gamma j$ using a Bayesian network with boosted training, ordered by the size of the amplitude. Figure from Ref. [31].

Closely related activities attempt to facilitate faster event unweighting and reweighting methods using NN generative models [24, 25] or fast to evaluate NN surrogates for the transition amplitudes [26].

## 2.2 Scattering Amplitudes

Perturbative precision calculations use interpolation methods to reduce the evaluation time for expensive loop amplitudes, defining a task where appropriately designed neural networks can be expected to outperform standard methods [26–30]. The challenge in NN-based surrogate models for integrands and amplitudes is to ensure that all relevant features are indeed encoded in the network at sufficient precision and to establish a reliable uncertainty treatment of the network training.

A relevant test case are loop-induced amplitudes such as those for

$$gg \to ZZ \qquad \text{and} \qquad gg \to \gamma\gamma + \text{jets}. \tag{4}$$

The application of simple, gradient boosted machines to $gg \to ZZ$ highlights that fast interpolation times can lead to significant improvements in overall simulation times, if reliable models

Table 2: Unweighting efficiencies for $V$+jets production at the LHC. 'SHERPA' relies on multi-channel importance sampling using VEGAS; 'NN' uses a normalizing flow; 'Gain' shows the improvement of NN over SHERPA. Results from Ref. [14].

| unweighting eff. $\epsilon_{\text{uw}}$ | | LO QCD | | | | | NLO QCD (RS) | |
|---|---|---|---|---|---|---|---|---|
| process/sampling | | $n=0$ | $n=1$ | $n=2$ | $n=3$ | $n=4$ | $n=0$ | $n=1$ |
| $W^+ + n$ jets | SHERPA | $2.8 \cdot 10^{-1}$ | $3.8 \cdot 10^{-2}$ | $7.5 \cdot 10^{-3}$ | $1.5 \cdot 10^{-3}$ | $8.3 \cdot 10^{-4}$ | $9.5 \cdot 10^{-2}$ | $4.5 \cdot 10^{-3}$ |
| | NN | $6.1 \cdot 10^{-1}$ | $1.2 \cdot 10^{-1}$ | $1.0 \cdot 10^{-3}$ | $1.8 \cdot 10^{-3}$ | $8.9 \cdot 10^{-4}$ | $1.6 \cdot 10^{-1}$ | $4.1 \cdot 10^{-3}$ |
| | Gain | 2.2 | 3.3 | 1.4 | 1.2 | 1.1 | 1.6 | 0.91 |
| $Z/\gamma^* + n$ jets | SHERPA | $3.1 \cdot 10^{-1}$ | $3.6 \cdot 10^{-2}$ | $1.5 \cdot 10^{-2}$ | $4.7 \cdot 10^{-3}$ | | $1.2 \cdot 10^{-1}$ | $5.3 \cdot 10^{-3}$ |
| | NN | $3.8 \cdot 10^{-1}$ | $1.0 \cdot 10^{-1}$ | $1.4 \cdot 10^{-2}$ | $2.4 \cdot 10^{-3}$ | | $1.8 \cdot 10^{-3}$ | $5.7 \cdot 10^{-3}$ |
| | Gain | 1.2 | 2.9 | 0.91 | 0.51 | | 1.5 | 1.1 |

can be trained on fewer points than the original Monte Carlos. To control the features of the amplitude relevant for differential cross sections, separating soft and collinear regions enables an ensemble of networks to reliably model full one-loop amplitudes for $e^+e^- \to\, \leq 4$ jets [28]. In Fig. 2 this scaling is shown for high-multiplicity scattering described by the NJET generator for $gg \to \gamma\gamma +$ jets. Simulations for hadron colliders show overall improvements of around a factor $N_{\text{inference}}/N_{\text{training}}$ [29] can be achieved. The right panel of Fig. 2 shows the achievable precision on the $\gamma\gamma j$ loop amplitude from a single Bayesian network with boosted training to improve the precision [32–35].

The reliability of the trained network is particularly at risk in divergent regions. However, these are precisely the phase space regions where the soft and collinear behavior of the amplitudes is universal and well known. Building the infrared factorization properties into the NN-based model can lead to substantial improvements for the tree-level $e^+e^- \to$ jets amplitude. Figure 3 shows that adopting a factorization-aware parametrization the achievable precision is brought down to the per-mille level for 5-jet production [30]. While the shown precision for this process does not translate into a clear improvement of higher-order LHC predictions, it illustrates how physics-informed network architectures can significantly improve the network precision as the key criterion for an application in the LHC simulation chain. Perhaps of even greater interest would be the use of a single trained model to integrate over a wide range of kinematic cuts, jet algorithms, PDF sets, scale choices, which could enable a further order of magnitude in overall performance.

## 2.3 Loop integrals

Amplitudes beyond the leading order contain loop integrals, and machine learning can improve the calculation of (multi-)loop integrals by optimizing the integrands in Feynman parameter space [36]. When an analytic solution to these integrals is not feasible, they must be evaluated numerically. Before attempting a numerical evaluation, the poles of the integrand need to be controlled. In dimensional regularization, ultraviolet and infrared poles can be factorized efficiently with sector decomposition. After factorizing these poles, integrable singularities related, for example, to thresholds, remain. Such poles, located on the real axis in Feynman parameter space, can be avoided by a deformation of the integration contour into the complex plane. An automated procedure to do this has already been implemented in standard tools like SECDEC, FIESTA, and pySECDEC. The deformation of the integration contour is not unique and can be performed in many ways. In fact, the numerical precision of the integration can vary by orders of magnitude depending on the chosen contour.

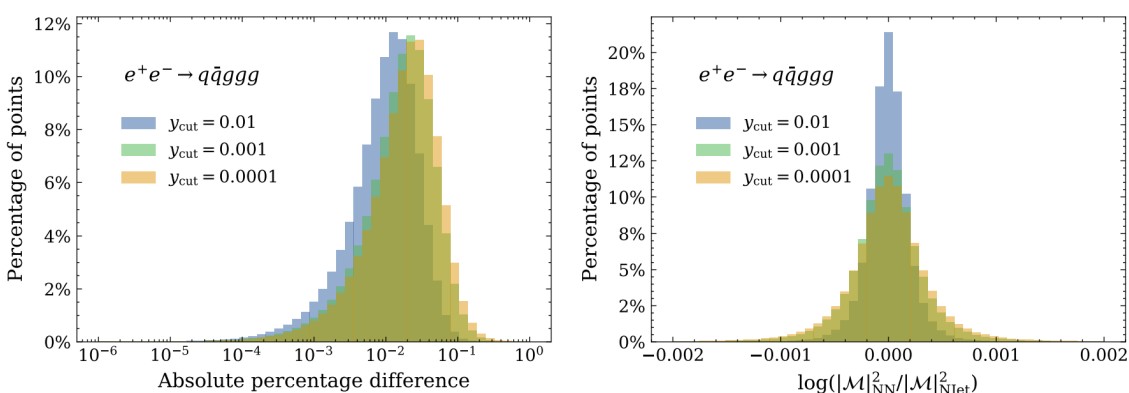

Figure 3: Accuracy for the full model from Ref. [30] for tree-level $e^+e^- \to 5$ jets amplitudes. Figure adapted from Ref. [30].

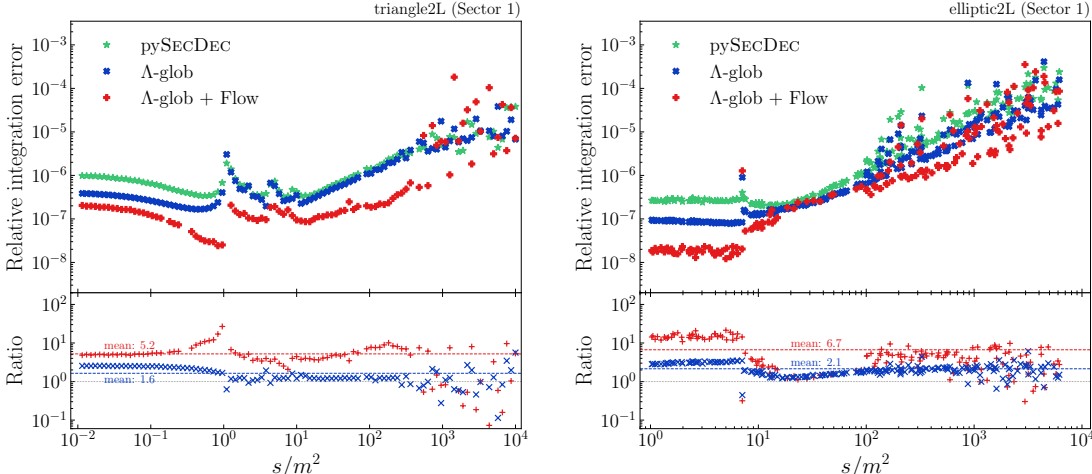

Figure 4: Relative integration error for sector one of a 2-loop triangle integral (left) and a 2-loop box integral known to contain elliptic functions (right) using the standard pySECDEC algorithm (green), the ML-assisted $\Lambda$-glob algorithm (blue) and including an additional normalizing flow (red). The lower panel shows the ratios to the standard method. The figures are taken from Ref. [36].

For standard integrals, the contour deformation procedure implemented in pySECDEC works fast and usually produces satisfactory contours in practice. However, for more complicated integrals and in specific phase-space regions, the chosen contour is sub-optimal and can be optimized significantly, see Fig. 4. In this case, ML-assisted, or more specifically, NN-assisted algorithms, offer great potential to amplify the precision. Like in the neural importance sampling methods [10, 13, 14, 17] for phase-space integrals, normalizing flows can be used to find a better parametrization of the integration domain. As these contour integrals need to satisfy certain boundary conditions, originating, for instance, from the Landau equations and Cauchy's theorem, the NN setup needs to be extended to obey these constraints. Furthermore, the usage of complex-valued floats can entail the necessity to construct own implementations for objects like gradients of complex determinants occurring during training and optimization.

## 2.4 Parton shower

The parton shower is an essential element of particle physics simulations. It describes the evolution of particles between the hard scale of the collision $\sim 100$ GeV to the hadronization scale $\Lambda_{\text{QCD}}$. This evolution is typically modeled as a Markov process where partons evolve semi-classically, radiating gluons as they move with probabilities determined only by properties of the parton splitting and perhaps one or two spectator partons in the event. Although the semi-classical approximation can be justified in the limit where the daughter particles are emitted at small angles with respect to the mother, parton showers are used well outside of this regime. The use of parton showers is thus justified not by physics but by necessity: computing the full distribution from first principles is computationally intractable. This limitation is an opportunity for machine learning; perhaps an improved parton shower could be learned rather than built.

The simulation of parton showers offers an interesting structure compared to other generative tasks. When simulating entire collision events, as discussed in Sec. 3, commonly a representation encoding a small and often fixed number of 4-vectors is chosen. Simulating showers all the way down to calorimeter sensors, or with calorimeter sensors themselves,

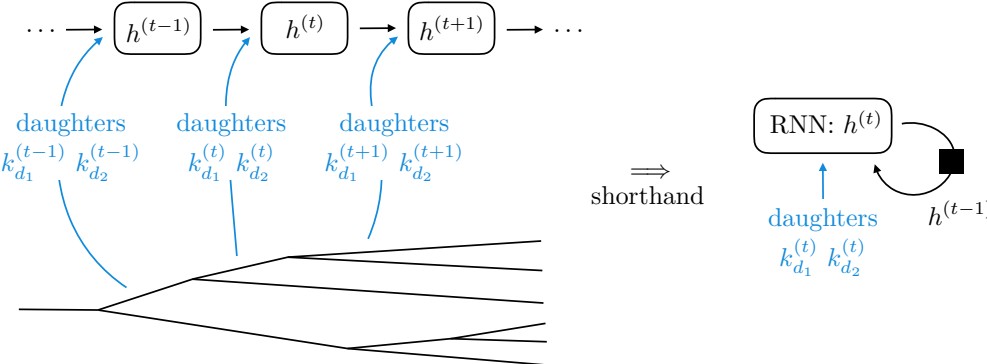

Figure 5: A promising approach to learning parton showers is to use a structure inspired by the semi-classical approximation as a backbone for a general probability estimator. In the JUNIPR approach, a recurrent neural network is used to emulated the Markov-process aspect of a parton shower. Figure taken from Ref. [37].

yields a much larger number of particles in the final state. However, the output nodes of a generative model can still be identified with different cells of the physical detector and therefore allow architectures that for example use convolutional layers.

Within the semi-classical approximation and even though the probability function at each branching in the shower is relatively simple, the overall distribution of particles produced is quite complex. It would be seriously challenging to learn this final distribution without some domain knowledge of its structure [38]. One approach is to scaffold a learnable model over a semi-classical framework [37,39], as sketched in Fig. 5. Additionally, network architectures based on sets or graphs explicitly encoding permutation symmetry of the final state particles have been investigated [40–45].

An alternative way of improving parton shower with ML-methods might be to stick to the fundamental splitting structure and measure the QCD splitting kernels in low-level observables. As before, the challenge of generating many particles covering several orders of magnitude in energy is taken care of by the usual Monte Carlo method. A modified and shower-specific form of the splitting kernels can be extracted from a combination of QCD predictions and data using ML-based inference [46]. While this approach has practical advantages, it is limited by the applicability of the simple splittings picture.

## 2.5 Parton distribution functions

Parton distribution functions (PDFs), encoding the structure of colliding protons, are vital for the calculation of hard scattering cross sections at the LHC and appear in several stages of the event simulation chain, in particular they guide the initial state parton showers and affect the underlying event activity. The determination of PDFs is a classic pattern recognition problem: it is known that an underlying law exists (the true analytic form of the PDFs, as determined by QCD in the non-perturbative regime) but its explicit form is not known, and it must be inferred from discrete data (the cross-sections of PDF-dependent hard processes), that moreover are correlated to it indirectly and in a convoluted way. In comparison to more standard pattern recognition problems, it has two peculiarities. First, the pattern – the set of PDFs — is a probability, rather than a deterministic outcome. Second, due to the noisy nature of the input, which is affected by both experimental and theoretical uncertainties, with a complex correlation pattern, the final deliverable is a probability distribution of possible results. Hence, one is delivering a probability distribution of probability distributions.

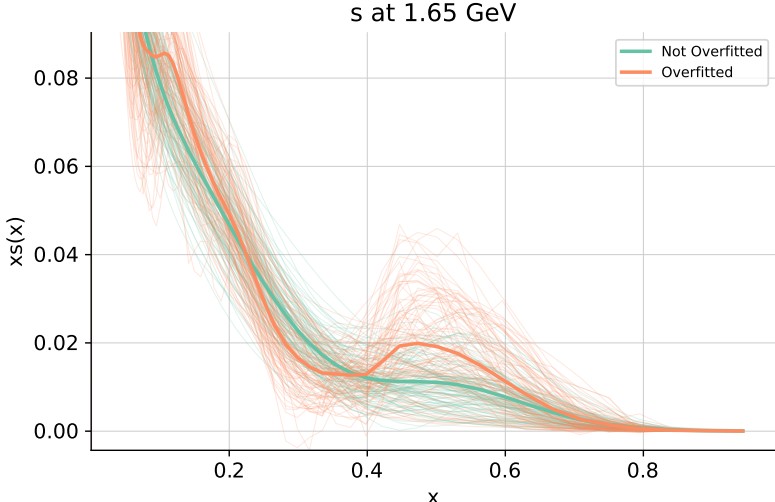

Figure 6: Comparison between two results for the strange-quark PDF: one where overfitting is clearly present, and one where this is not the case. Figure taken from Ref. [47].

The way to approach this problem as a machine learning challenge was first suggested long ago [48]: the basic idea is to deliver a Monte Carlo ensemble of machine learning models, such as neural networks, that provide the desired representation of a probability of probabilities. The successful implementation of this idea has led to the NNPDF family of proton PDF determinations [47, 49–51] as well as to variants in the context of polarised PDF [52] and nuclear PDF [53, 54] global analyses. The current implementation frontier, which has led to the recent NNPDF4.0 determination, involves a suite of contemporary machine learning methods and tools, specifically cross-validation to avoid overtraining, hyperoptimization [55] combined with $K$-folding for the automatic selection of the methodology, feature scaling of the input for the optimization of the neural networks used as basic underlying model [56], and GAN-enhanced compression for final efficient delivery [57, 58].

The current main challenge remains the maximal optimization of the extraction of available information while avoiding overfitting, and the generalization to cases in which information is scarce or altogether absent, such as extrapolation to kinematic regions where there are no data. This is the physically most interesting case, as these are the regions where new physics is being searched for, and also a challenge at the frontier of machine learning. While several machine learning tools have been implemented with the aim of preventing overfitting, confirming whether the PDF resulting of a fit is indeed free of overfitting still relies – at least in part – on the fitter's accumulated knowledge of PDFs. To illustrate this point, Fig. 6 shows a comparison of the strange-quark PDF $xs(x, Q)$ at $Q = 1.65$ GeV, both for a good fit and a clearly overfitted alternative. The development of reliable quantitative measures of the degree of overfitting is a challenge, both within the context of PDF determination and more in general in machine learning, and it is a topic of ongoing research.

## 2.6 Fragmentation functions

Fragmentation functions (FFs) are the time-like equivalent of PDFs and encode the probabilities associated to the transition between partons produced in the hard-scattering and specific types of hadrons. Being based on the perturbative QCD factorization framework, FFs represent an alternative strategy to model partonic hadronization as compared to the phenomenological models available in most MC event generators. FFs can be determined from a global analysis of

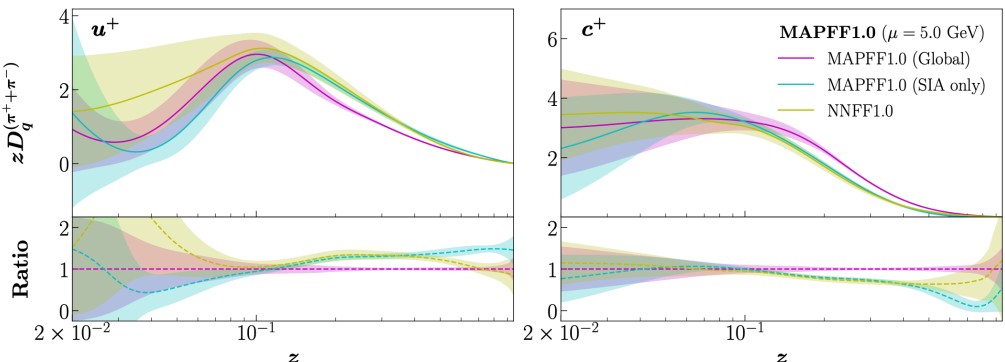

Figure 7: Fragmentation functions of up (left) and charm quarks (right) into charged pions as a function of the time-like momentum fraction $z$, comparing the results of two approaches (MAP and NNFF) based on machine learning techniques. Figure taken from Ref. [59].

hard-scattering data from electron-positron collisions, semi-inclusive deep inelastic scattering, and proton-proton collisions (RHIC and LHC) with identified final-state hadrons.

A phenomenological analysis of FFs requires introducing a parametrization for their initial-scale ($Q_0$) dependence with the momentum fraction $z$, $zD_i^{(h)}(z,Q_0)$, where $i$ is a partonic index and $(h)$ a hadronic label. To remove theory bias and model-dependence in the determination of FFs, machine learning techniques can be adopted [59–62]. Feed-forward neural networks are deployed as universal unbiased interpolants for $zD_i^{(h)}(z,Q_0)$, whose weight and threshold parameters are obtained from a log-likelihood maximization by comparison with experimental data. This approach can be combined with the Monte Carlo replica method, originally deployed for PDFs [63], to estimate and propagate the uncertainties from the input data to the output FFs. The basic strategy is to generate $N_{rep}$ replicas which sample the probability density associated to the data, and then train a separate neural network to each of these replicas. The spread of the resulting networks (i.e. 68% CL intervals) provides then a robust estimate of the uncertainties associated to the FFs.

Fig. 7 displays a comparison between FFs determined in two approaches (MAP and NNFF) based on machine learning techniques. We show the FFs associated with the transition of up and charm quarks into charged pions ($\pi^+ + \pi^-$) as a function of the time-like momentum fraction $z$. The bands represent the corresponding 68% CL ranges. It is worth emphasizing that the resulting shapes, given the outcome of the NNs, are completely driven by the data, with no specific models (more or less inspired by QCD) assumed. The combination of the FFs $zD_i^{(h)}(z,Q_0)$ obtained in this manner with higher-order perturbative QCD calculations provides precise and accurate predictions for hard-scattering processes including identified hadrons in the final state, which are important for many key phenomenological applications.

## 3 End-to-end ML-generators

In addition to applying a wide range of machine learning tools to improve the modules of classic event generators, we can train generative neural networks to directly generate events, at parton level and with or without detector effects [7]. End-to-end or, better, soup-to-nuts ML-generators have to be developed together with the established generators and serve as studies for phase space generators, enable inverted simulations, provide datasets for phenomenological analyses, and allow us to efficiently ship event samples. Their advantages include training

on data combined with simulations, manipulation of event samples [64], or post-processing of MC data for example to unweight events [17, 24, 26]. Finally, they define useful benchmarks for conceptual work on uncertainty estimates for generative neural networks.

The work horses behind ML-generators are GANs, VAEs, optimal-transport-based probabilistic autoencoders, normalizing flows, and their invertible network (INN) variant. Given the interpolation properties of neural networks and the benefits of their implicit bias in the applications described in Sec. 2, we can quantify the amplification of statistics-limited training data through generative networks [65, 66].

## 3.1 Fast generative networks

Theory-driven ML-generators at the parton level [18, 69] can be combined with experiment-driven fast detector simulations [70–81] into single generative networks [68, 82–86], provided we have sufficient control over the network and its uncertainties. Single, soup-to-nuts simulation networks are inspired by the fundamental goal of the detection process, namely to reconstruct parton-level information as accurately as possible.

Comparing different generative network architectures, we start with highly expressive VAEs. They can be trained to generate events at the parton level, without or with fast detector simulation, by maximizing a lower bound of the data likelihood through variational inference (ELBO). The model consists of a decoder $p(x|z)$ which maps from a latent space $\mathcal{Z}$ to the phase space $\mathcal{X}$, and an encoder $q(z|x)$ which is a variational approximation to the inverse of $p(x|z)$. In practice, it is difficult to simultaneously optimize the separate components of the ELBO and the VAE performance can be improved by weighting the KL-divergence in the loss function term by a factor $\beta$. The B-VAE [20] is characterized by the limit $\beta \ll 1$ and a strong preference for the reconstruction loss. After optimization, the Gaussian latent distribution is replaced by a buffer which consists of the latent distribution derived from training events. This model simultaneously achieves a highly-optimized reconstruction loss, but with a closely-matched and non-Gaussian latent distribution. While VAEs are very expressive probabilistic models, the approximate nature of the ELBO and the need to balance the two components of the loss function can become limiting factors.

Similarly, GANs can extract and reproduce the phase space density of LHC events. While

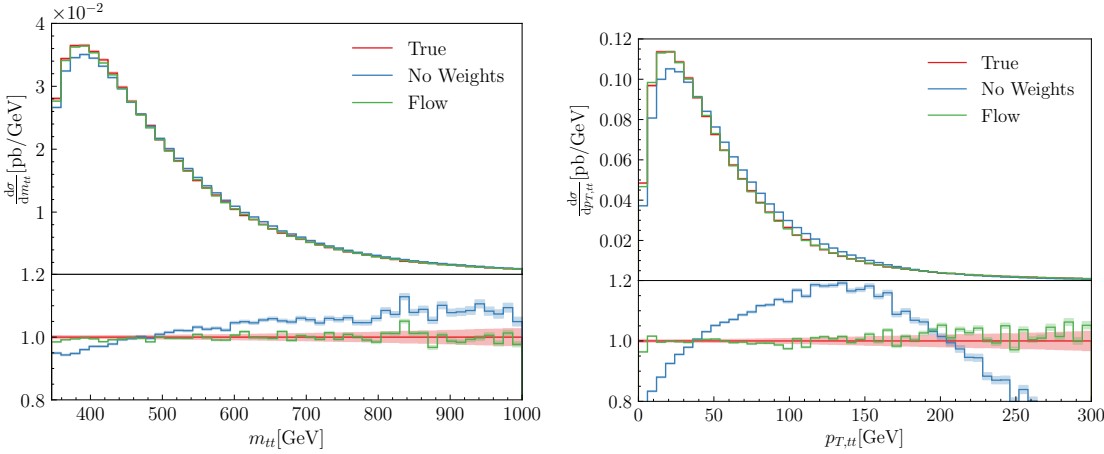

Figure 8: Distributions of the invariant mass (left) and transverse momentum (right) of the $t\bar{t}$ system in $pp \to t\bar{t}$ generated using MC@NLO. The true distribution (red) is compared with the normalizing flow distributions excluding (blue) or including (green) negative event weights. Figure from Ref. [67].

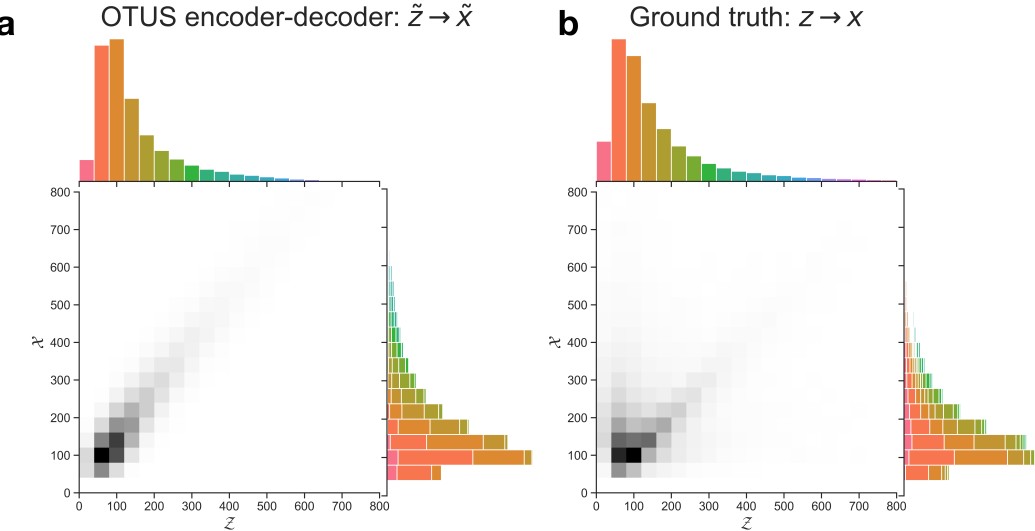

Figure 9: Visualization of the transformation from parton-level theory-space to reconstructed data-space, $\mathcal{Z} \to \mathcal{X}$. Left: learned transformation of OTUS's decoder, $p_D(x \mid z)$. Right: true transformation from simulated samples, for comparison. Colors in the $\mathcal{X}$ projection indicate the source bin in $\mathcal{Z}$ for a given sample. Figure from Ref. [68].

technically the difference between training on events without or with detector effects is negligible, parton-level events are more challenging when it comes to sharp kinematic features like Breit-Wigner mass peaks. GANs generically do not achieve the necessary precision for such features, so they have to be enhanced, for example with a targeted MMD loss [18]. The main challenge of GANs is the precision they can achieve in the underlying phase space density while finding a Nash equilibrium.

Finally, normalizing flows avoid some of the limitations of the above architectures for LHC event generation [67, 69, 87]. At the cost of some flexibility, they offer a direct evaluation of the likelihood without having to resort to variational inference. They start from a latent space $\mathcal{Z}$ and apply a series of bijective transforms, with tractable Jacobian, to the phase space $\mathcal{X}$. While the expressivity of the model may in some cases be limited, the advantage of a tractable likelihood is significant. Flows can be trained on weighted events, including negative weights, through a simple modification of the loss [67]. Figure 8 illustrates their performance applied to $pp \to t\bar{t}$ events at the parton level, including shower evolution, generated with MC@NLO. In addition, normalizing flows come with significant advantages in controlling their performance and quantifying uncertainties, as discussed in the next section. Their invertible structure is useful for many LHC-applications, including anomaly detection or related density estimation tasks [88–91].

An attractive application of soup-to-nuts networks can be targeted using Optimal Transport-based probabilistic autoencoders [68]. Their structural advantage is that they learn the mapping from parton-level information in theory space, $\mathcal{Z}$, to detected and reconstructed objects in data space, $\mathcal{X}$, without requiring paired event samples, $\{z, x\}$. The probabilistic autoencoder's latent space is identified with a physically meaningful representation of parton-level theory-space information, so the encoder and decoder networks define a simulator mapping, $\mathcal{Z} \to \mathcal{X}$, and an unfolding mapping, $\mathcal{X} \to \mathcal{Z}$. Properties of the OT-based method encourage the encoder and decoder to be conditional mappings, effectively sampling from the probability distributions $p_E(z|x)$ and $p_D(x|z)$, respectively. Over many samples, these distributions will marginalize to the appropriate theory-space and data-space priors, $p(z)$ and $p(x)$,

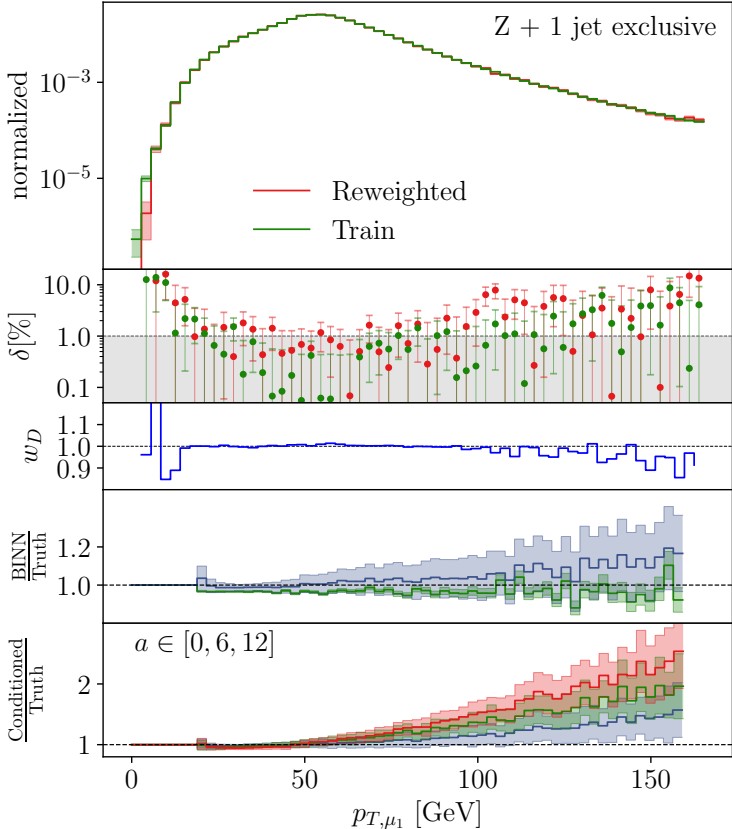

Figure 10: Illustration of a complete control and uncertainty treatment for generative networks applied to LHC event generation and simulation. Figure from Ref. [69].

respectively. Alternative methods to encode an unfolding mapping in neural networks are discussed in Sec. 4.

Despite having no training pair information, OTUS's learned mappings exhibit physical-behavior, even picking up on invariant masses which were withheld during training. This suggests that further development in this direction should produce physically meaningful mappings, even if relations are missed or unknown, and therefore not included in the training process. On the other hand, providing known relations as inductive biases on the data inputs, network architectures, or loss functions will likely improve performance. Figure 9 depicts the joint distribution and marginals of OTUS' trained simulator as well as the true simulator for one test-case. Despite OTUS only having information about marginal-matching during training, the decoder network learns a mapping which is qualitatively similar to the true simulator.

## 3.2 Control and precision

If we use neural networks to encode theory predictions for the LHC, we need to ensure that all relevant phase space features are described with the required precision [92]. For neural networks, this problem can be split into two distinct tasks: first, we need control over the relevant phase space features, so the network does not interpolate over relevant, but narrow phase space regions. Second, we have to estimate the precision with which the network has learned the underlying phase space density.

Neural networks work much like a fit and not like an interpolation in the sense that they do not reproduce the training data faithfully and instead learn a smooth approximation [65, 66]. This is where we can gain some intuition for a NN-uncertainty treatment. For a fit,

uncertainties on the training data are crucial information in the loss function. We then monitor the fit quality and ensure that the fit is reliable over the entire phase space.

To guarantee that all relevant features are encoded in a generative network, we can follow the GAN inspiration and train a simple discriminator network to identify and quantify deviations between training and generated data. As a post-processing step such a discriminator can be used to reweight the events from the generative network [25,69,93]. In the GAN spirit we can incorporate the discriminator into the generator training, either through adversarial training searching for a Nash equilibrium, or through alternative approaches for a normalizing flow generator. Such a joint training will improve the generator, provide an uncertainty estimate, and prepare any remaining information in the discriminator for reweighting, as illustrated for $Z$+jets production at the parton level in Fig. 10.

Once we know that the neural network describes all features, we determine how well it does. This can be done with Bayesian networks, where the learned network weights are replaced by learned network weight distributions [32,33]. Bayesian network approaches have been shown to describe uncertainties in regression [35] and classification [34] tasks, and the concept can be expanded to generative networks [87]. For generative networks we can assign a training-related uncertainty in the underlying phase space density to the (unit) weight of each event. In Fig. 10 we see, for instance, the increasing uncertainty in the kinematic tail, driven by a lack of training data.

We often know systematic or theoretical limitations of describing certain kinematic regimes. In that case we augment the training data, representing this uncertainty through an additional parameter in event weights. We train the generative network conditionally on this parameter, either in a deterministic or a Bayesian setup, and generate events either for a given parameter or sampling over it. Again, this approach is illustrated in Fig. 10, where $a$ directly affects the $p_T$-distribution of the leading jet and enters many other observables through kinematic correlations. We see that its effect is larger than the uncertainty from the Bayesian network for the individual $a$-values. This first attempt of a comprehensive uncertainty treatment for generative networks will allow us to build confidence in the applications of generative networks to LHC simulation and inference.

## 4 Inverse simulations and inference

Monte Carlo simulations based on first principles have allowed us to properly understand essentially all aspects of LHC data. The price to pay for an extremely fast and reliable forward Monte Carlo simulation chain is that the corresponding inverse simulation is not feasible in practice. ML-based simulations can be built symmetrically, for instance INNs encode a bijective mapping between two physics spaces linking different levels of the simulation chain illustrated in Fig. 1 [94, 95]. Similarly, we can relate different levels of the simulation chain through a reweighting procedure working on the full respective phase spaces and accounting for all correlations [96]. Moreover, as ML-based simulations are often differentiable, we can use their gradients to probe and learn about distributions on phase space [97]. Finally, we can construct generative inverse simulations with conditional versions of the respective forward generative networks [95, 98, 99]. This last approach is based on progress with soup-to-nuts ML-generators and their essentially identical network architectures.

### 4.1 Particle reconstruction

The first stage of the inverse problem uses the set of energy deposits in the detector to reconstruct the set of particles present at the first interaction with the detector, that is, following hadronization. In its fullest sense, reconstruction also involves the prediction of the particles'

properties, in particular, their class and momenta. The difficulty of this task stems from the busy LHC environment caused by pileup interactions and the inherently collimated signatures associated with jets. Traditional particle flow (PF) algorithms rely on parameterized schemes for merging and splitting to disentangle overlapping calorimeter cell clusters as well as track-based subtraction to infer the contribution from neutral particles.

A series of publications [100–102] have established the potential for ML-based reconstruction to go beyond traditional PF algorithms. In Ref. [100], particle reconstruction was recast as a computer vision problem using state-of-the-art ML architectures including U-net, graph neural network (GNN) and DeepSets. A simplified dataset was used comprising overlapping pairs of charged and neutral pions in a 6-layer calorimeter block. In comparison to a traditional PF algorithm, the ML models regress the component of neutral energy better by a factor of two to four in terms of resolution. The study also finds significant improvements via a super-resolution approach (see also Ref. [103]), where the network is trained to predict a corresponding calorimeter signature with higher granularity.

This proof of concept has been extended to particle reconstruction in more realistic environments resembling multiple pileup interactions in a full-coverage simulated detector [101,102]. In both cases, GNN architectures are employed for their ability to handle the complexity of detector data: variable numbers of input and target entities, lack of ordering, irregularity of detector components, and sparsity of "pixels". Moreover, GNNs are able to leverage the spatial relationships between calorimeter cells alongside their input features to optimize the prediction tasks.

Based on these developments, it can already be anticipated that ML methods will take a key role in particle reconstruction at future runs of the LHC, especially to handle HL-LHC conditions. GNN-based models in particular show potential to outperform current PF algorithms for particle identification and regression while opening new possibilities such as super-resolution and resolving neutral particles inside of jets. Finally, the learned deep latent representation of detector information, which underlies the prediction tasks, should serve as a more expressive input format for both event classification and downstream tasks in the inverse problem.

## 4.2 Detector unfolding

While the physical processes behind an LHC collision are described by fundamental physics and are therefore universal, the observed data depend in an intimate way on the technical details of the detector. Detector effects like phase space coverage, detection thresholds, particle reconstruction, efficiencies, or calibration induce not only resolution smearing in the measurements, but can lead to systematic deviations between the properties of particles reaching the detector and the objects reconstructed from actually measured data. For individual experiments, these detector effects differ greatly and can only be estimated by the collaboration. It is therefore essential for future interpretations of a measurement to unfold detector effects so that we can compare measurements by different experiments to each other and to theory predictions.

Traditional approaches to unfolding are based on matrices connecting binned particle-level distributions at truth level with histograms of corresponding detector-level observables. While the folding or convolution of detector effects with kinematic distributions at particle level is possible with Monte Carlo simulations, the inverse direction often suffers from instabilities and scales poorly for high-dimensional phase spaces. The limitation to low-dimensional representations requires an unwanted pre-selection of interesting observables. Finally, the matrix-based approach requires fixed bin sizes, which limits the re-optimization options for future analyses.

ML-approaches establish high-dimensional and binning-independent unfolding. We can distinguish two fundamentally different concepts [104]: a classification-based approach to reweight a Monte Carlo simulation with the learned likelihood ratio of data and simula-

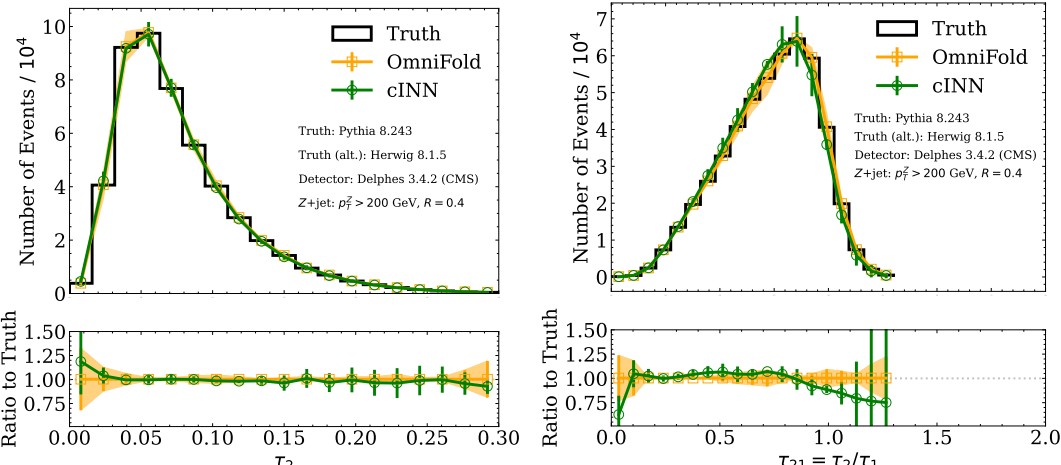

Figure 11: An illustration of classifier-based (OmniFold) and density-based (cINN) unfolding for the two-dimensional space of $N$-subjettiness variables $\tau_1$ and $\tau_2$. The right plot shows that since the unfolding is done simultaneously and unbinned, we can produce a measurement of the widely-used $N$-subjettiness ratio for free. Figure from Ref. [104].

tion [96]; and a complementary approach that learns directly the probability density at particle-level [95]. Figure 11 illustrates that both approaches can perform unbinned unfolding in multiple dimensions.

Classification-based approaches start by learning the likelihood ratio between data and simulation at reconstruction level [96, 105, 106]. Using matched event pairs at the truth and reconstruction levels, the resulting weights are pulled to the particle level. Next, a classifier learns the likelihood ratio of the weighted and unweighted distributions at particle level to replace an event-based weight with a generalized weighting function. After several iterations of weight updates, the algorithm converges to an unfolded distribution which is compatible with the observed measurement.

Density-based approaches build on generative networks that predict probability configurations of truth-level events given a detector-level measurement. They are trained on pairs of reconstruction- and particle-level events from Monte Carlo simulations, to learn a direct mapping between both levels. Unfolding built on generative adversarial networks has been shown to work on kinematic distributions [94, 99]. Event-wise unfolding requires a meaningful probabilistic treatment, which can be achieved with conditional normalizing flows [95, 97], the kind of generative networks which also allows for the uncertainty treatment discussed in Sec. 3.2. This unfolding method yields calibrated probability distributions for each measured event. It admits multiple approaches; one approach frames unfolding in terms of learning a conditional density of particle-level quantities conditioned on reconstructed inputs [95], while another approach frames unfolding as an empirical Bayes / maximum marginal likelihood / data-informed prior learning problem [97].

Because classification-based and density-based unfolding techniques have distinct strengths and weaknesses, the natural next step will be to combine the two methods to benefit from both. While there is an extensive R&D program required to integrate both methodologies and to achieve precision, these tools are starting to be applied to data analysis in collider physics [107]. Looking ahead, it is clear that future versions of these tools will play an important role in the data analysis of future colliders. Unfolded differential cross sections are one of the main data products from collider experiments. By performing the unfolding with as much information as possible, we ensure that the measurements achieve the maximal preci-

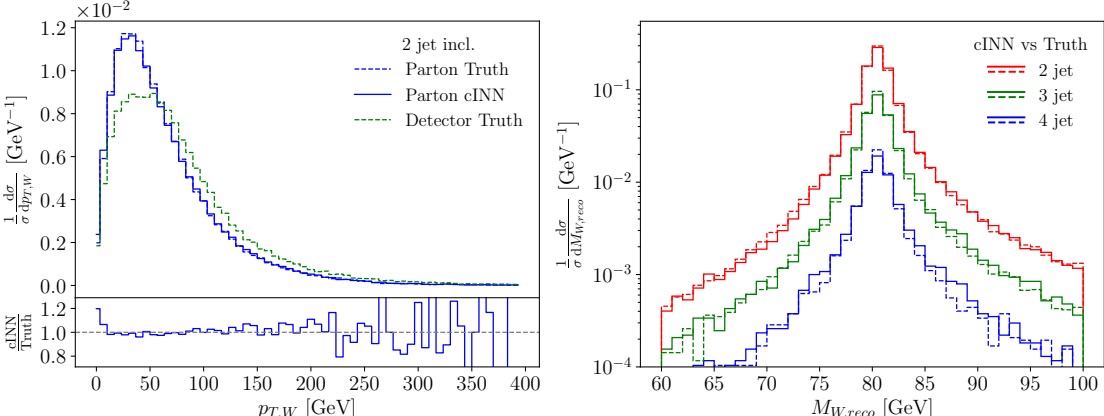

Figure 12: Unfolded parton-level distributions for the process $pp \rightarrow Z_{\ell\ell}W_{jj}$+jets using a cINN. The unfolding covers detector effects as well as additional jet radiation. Figure from Ref. [95].

sion, making the best use of the data. Furthermore, high-dimensional and unbinned unfolding ensures that these data products are 'future proof' in the sense that binning and even observables can be chosen post-hoc [104]. This enables downstream data analysis long after the data were published, including when new theoretical insights are available.

## 4.3 Unfolding to parton level

Once we control ML-unfolding of detector effects, we can target other parts of the simulation chain shown in Fig. 1 and invert them for a given LHC analysis. To probe the kinematics of a hard scattering process we can use neural networks to unfold QCD jet radiation and heavy particle decays to study the production kinematics of top quarks, electroweak gauge bosons, or the Higgs without binning and with full correlations. Such measurements are standard in top physics and provide the ideal input to global SMEFT analyses. Once we know the parton-level configurations for a given observed event, we can use NN-techniques to evaluate observables like CP-sensitive angular correlations in their original reference frames.

The inversion of QCD radiation or decays relies on the same classification or generative networks as detector unfolding. For instance, we can train a normalizing flow to map random numbers to the parton-level phase space, under the condition of a given detector-level event. The underlying model is encoded in the forward simulation chain used to train the network. Part of it is the assumed hard process, including the number of jets which are part of the hard scattering and do not get unfolded. When analyzing an event and sampling into parton-level phase space, we extract a probability distribution of parton-level configurations [95], which we can use to define observables suitable for standard analyses.

One challenge for such analyses are combinatorics. For the hard scattering $q\bar{q} \rightarrow Z_{\ell\ell}W_{jj}$ and up to two additional QCD jets we ask how well cINN-unfolding extracts the $W$-kinematics. In the left panel of Fig. 12 we illustrate how the network reproduces the momentum of the decaying $W$-boson. The relation between the up to four jets and the two partonic quarks from the $W$-decay is learned by the network. In the right panel of Fig. 12 we show the reconstructed $W$-mass stacked for different numbers of jets. The network resolves the underlying combinatorics such that the $W$-widths for the different jet multiplicities are identical, all by by accessing correlations combined with the truth information from the forward simulation. This corresponds to results from a systematic study which shows that deep networks outperform classical approaches to solving the combinatorics in the reconstruction of top-quark final states significantly [108].

High-level observables encoded into neural networks will find their way into standard experimental analyses. They are motivated by existing top-sector measurements, and using NN-techniques will simplify their use considerably. Moreover, the comprehensive uncertainty treatment discussed Sec. 3.2 and the merged classification-based [96] and density-based techniques from Sec. 4.2 can be applied to any part of an inverted or unfolded simulation chain.

## 4.4 MadMiner

The relation between data $x$ and physics parameters $\theta$ is, fundamentally, described by the likelihood function or normalized fully differential cross section, which we can predict in a factorized form,

$$p(x|\theta) = \frac{1}{\sigma(\theta)} \frac{d\sigma(x|\theta)}{dx}. \tag{5}$$

While we can predict this likelihood at the detector level using the standard, forward simulation tools, we can only compute it in a closed form at the parton level. This challenge in the relation of simulations and inference is where neural networks might lead to transformative progress.

Inspired by the standard simulation chain we can assume that the likelihood of Eq.(5) approximately factorizes into the form [109, 110]

$$p(x|\theta) = \int dz_d \int dz_s \int dz_p \underbrace{p(x|z_d)\,p(z_d|z_s)\,p(z_s|z_p)\,p(z_p|\theta)}_{p(x,z|\theta)}. \tag{6}$$

Here we integrate over latent variables $z$, where $z_d$ characterize the detector effects, $z_s$ the parton shower and hadronization, and $z_p$ the partonic phase space including helicities, charges, and flavors, etc. Given the typically large number of latent variables, it is unrealistic to integrate over them or evaluate the joint-likelihood $p(x, z|\theta)$. However, it is possible to calculate the joint likelihood ratio relative to a reference point in terms of the ratio of squared matrix elements from parton-level generators [109–113],

$$r(x, z|\theta) = \frac{p(x, z|\theta)}{p(x, z|\theta_{\rm ref})} = \frac{p(z_p|\theta)}{p(z_p|\theta_{\rm ref})} \sim \frac{|\mathcal{M}|^2(z_p|\theta)}{|\mathcal{M}|^2(z_p|\theta_{\rm ref})} \frac{\sigma(\theta_{\rm ref})}{\sigma(\theta)}. \tag{7}$$

The starting point to new ML-methods is to construct functionals in terms of the joint likelihood ratio $r(x, z|\theta)$, which are minimized by the true likelihood or likelihood ratio function [114, 115]. The result of this training are neural networks that approximate the true likelihood ratio $r(x|\theta)$. Given such a neural network, established statistical techniques can be used to construct confidence limits in parameter space.

Note that here simulation-based inference provides the primary statistical model, i.e. the probability model $p(x|\theta, \nu)$ that describes the dependence on the data $x$, the parameters of interest $\theta$, and the nuisance parameters $\nu$, even when the data is high-dimensional and traditional modeling approaches are inadequate. The publication of the trained network in a re-usable form, as discussed below, can thus be of great benefit for an optimal use of experimental results [117]. This approach is separate from tools like DNNLikelikood [118], which aims at approximating likelihoods derived from traditional approaches to model building, using libraries like RooFit.

Instead of the full likelihood function, one can also use the score $t(x|\theta) = \nabla_\theta \log p(x|\theta)$ to define statistically optimal observables at the detector level. This approach is motivated by an expansion of the log likelihood ratio around $\theta_{\rm ref}$,

$$\log r(x|\theta) = \log r(x|\theta_{\rm ref}) + t(x|\theta_{\rm ref})\,(\theta - \theta_{\rm ref}) + \cdots. \tag{8}$$

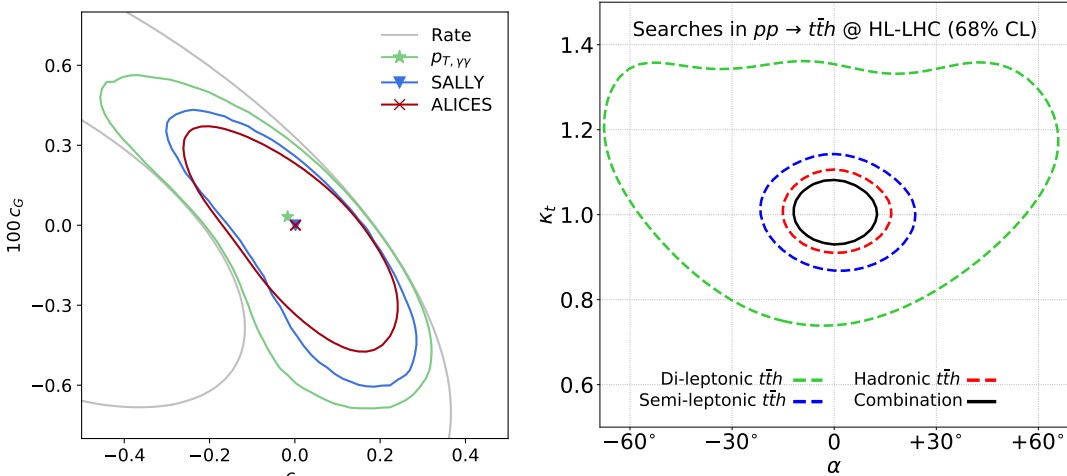

Figure 13: Expected sensitivity of a MadMiner based analysis for $t\bar{t}H$ production, probing SMEFT coefficients (left) and the CP-structure of the top Yukawa coupling (right). Figures taken from Ref. [113] and Ref. [116], respectively.

For parameter points close enough to $\theta_{\text{ref}}$ the score components are the sufficient statistics, so for measuring $\theta$ knowing $t(x|\theta_{\text{ref}})$ is as powerful as the full likelihood. Since the score is defined through the likelihood function, it is also intractable. However, similarly to the approach discussed above, we can train a neural network on a suitable loss function such that it will converge to the score. The trained network will now represent the optimal observable. In a next step, the likelihood can be determined for instance with simple histograms of the score components [113, 119]. This approach requires only minor changes to established analysis pipelines. Alternatively, the scores can be used to evaluate the Fisher Information and set limits based on the Cramer-Rao bound [113]. One challenge with training using the score is that the relevant gradient information of the matrix elements must be accessible for training the neural network, but this information is typically only accessible for a subset of parameters with analytic dependence that facilitates easy gradient estimation. One approach to enable score based training for any parameter is through differentiable programming; when matrix elements are merged with automatic differentiation frameworks, the required gradients can be computed automatically with relatively small additional computational overhead. Case studies using differentiable matrix elements from MADJAX for score based training successfully trained networks for inference on parameters that were inaccessible without differentiable matrix elements [23].

The previously outlined inference strategy has been fully automated in the MadMiner tool [113, 120]. The increase in physics sensitivity relative to a total rate or single kinematic distribution is illustrated in Fig. 13. In the left panel we consider $t\bar{t}H$ production to constrain the two SMEFT Wilson coefficients $c_u$ and $c_G$. In the right panel we consider the same process to constrain CP-violation in the top-Higgs coupling, as parameterized by the magnitude $\kappa_t$ and CP-phase $\alpha$ of the top Yukawa coupling [116, 121].

## 4.5 Matrix element method

Inverting the entire simulation chain in Fig. 1 allows us to extract the transition amplitude for an observed event and relate it to the theory prediction. This so-called matrix element method (MEM) can be used to estimate fundamental physics parameters from individual events and has, for instance, been applied for measure the top mass. Being defined on the single-event level, it is in particular suitable for low-statistics signals, where an optimal exploitation of all

kinematic features is critical.

The MEM relies on our ability to extract the likelihood for detector-level events as a function of a model parameter $\theta$, as an ML-application through density-based unfolding or an inverted simulation. Extending the discussion in Sec. 4.4, the transition amplitude as a function of detector-level phase space is unknown, but it can be calculated at the parton level. The two phase spaces can be related by transfer functions $T(\vec{x}, \vec{z})$, probabilities to observe parton-level configurations $\vec{z}$ as detector-level signatures $\vec{x}$, as part of the forward simulation. In the $N$-event likelihood they appear as

$$\mathcal{L}(\theta) = \prod_{i=1}^{N} p(\vec{x}^{(i)}|\theta) = \prod_{i=1}^{N} \frac{1}{\sigma_{\text{fid}}(\theta)} \left.\frac{\mathrm{d}^l \sigma(\theta)}{\mathrm{d}x_1 ... \mathrm{d}x_l}\right|_{\vec{x}^{(i)}} = \prod_{i=1}^{N} \frac{1}{\sigma_{\text{fid}}(\theta)} \int \mathrm{d}^m z \, \frac{\mathrm{d}^m \sigma(\theta)}{\mathrm{d}z_1 ... \mathrm{d}z_m} \, T(\vec{x}^{(i)}, \vec{z}). \quad (9)$$

The dimensionality of the parton-level and detector-level phase spaces is different. For instance, longitudinal neutrino momenta are unobservable, while additional jets have to be included with higher-order QCD corrections. Existing approaches model the transfer functions heuristically, and for non-trivial cases the numerical convolution is impossible. The form of Eq.(9) indicates ways of enhancing the accuracy of the matrix element method: first, higher-order corrections can be included at parton level, for instance using the MEM@NLO program. Second, general and highly non-Gaussian transfer functions can account for parton shower, hadronization, detector resolution, acceptance, and efficiency, as well as a possible mismatch between theoretically described and actually measured quantities, event by event.

The transfer function is defined as a probability density $T(\vec{x}, \vec{z}) = p(\vec{x}|\vec{z}, \theta)$. This allows us to learn it directly from simulated data using a conditional normalizing flow or INN as a density estimator. Because the matrix element spans several orders of magnitude and the transfer function usually is a narrow peak in phase space, the integral in Eq.(9) is numerically challenging for a regular Monte-Carlo integration. However, we know from Bayes' theorem that the integrand becomes trivial when the parton level samples are drawn from the distribution $p(\vec{z}|\vec{x}, \theta)$. In analogy to Secs. 4.2 and 4.3, the partonic configurations $\vec{z}$ for a given detector event $\vec{x}^{(i)}$ can be sampled by another conditional INN. Then the likelihood can be expressed as

$$\mathcal{L}(\theta) = \prod_{i=1}^{N} \frac{1}{\sigma_{\text{fid}}(\theta)} \left\langle \frac{\partial \vec{z}(\vec{r}; \vec{x}^{(i)}, \theta)}{\partial \vec{r}} \left[ \frac{\mathrm{d}^m \sigma(\theta)}{\mathrm{d}z_1 ... \mathrm{d}z_m} T(\vec{x}^{(i)}, \vec{z}) \right]_{\vec{z}(\vec{r}; \vec{x}^{(i)}, \theta)} \right\rangle_{\vec{r} \sim p(\vec{r})}. \quad (10)$$

This way, density estimation of the transfer function in combination with density-based unfolding will allow us to make optimal use of the statistical power of the MEM, exploiting the full and correlated event kinematics event by event for critical LHC observables like the top mass, the Higgs self-coupling, or CP-violating phases.

# 5 Synergies, transparency and reproducibility

A key paradigm in the development of simulation tools for high-energy collider experiments is publicly accessible open source software. The versioning of code releases and the reproducibility of predictions is vital for a reliable analysis and interpretation of collider data. As we have seen in the previous sections, ML-methods are entering all aspects of the simulations chain at high pace. They range from initial proof-of-concept applications to well established use cases with largely consolidated techniques, for example in the determination of parton densities.

Machine learning models efficiently encode arbitrary decision functions of a given set of inputs, and thus offer a chance to easily exchange complex relations. This might correspond to the value of a scattering matrix element given a set of momenta, or probability models

from simulation-based inference, like MEM or MadMiner. The sharing of neural networks used for various generative or discriminative tasks will be of central importance and should be further extended. This will allow researchers to critically examine and build upon previous results more easily, enable synergies between different use cases, and facilitate reproducibility of results.

Successfully sharing a machine learning model entails two challenges: (i) sharing the model itself, including architectures, software versions, and weights; and (ii) sharing data it can be used on. Exchanging models is technically relatively straightforward and several corresponding tools exist, for example Open Neural Network Exchange (ONNX). It allows the exchange of neural networks and BDTs between training frameworks.

Suitable input data poses the more difficult problem. On the side of results by large collaborations, this adds additional weight to the ongoing move towards publishing open data along with measurements. Containerization, as enabled by software tools like Docker can be useful in bundling the correct versions of different software packages used for data processing and machine learning in a coherent fashion.

An opportunity exists in the realm of phenomenological studies based on the DELPHES detector simulation [122]. Here a common specification on how quantities are translated into the inputs to machine learning algorithms might — together with publishing the ML models — boost sharing and meaningful exchange. Another interesting angle are generative models. As these do *not* need data to evaluate, sharing the architecture and weights is already sufficient. Generative networks themselves can even be used as an efficient alternative way of sharing simulated data.

Publication of ML models for their reuse is not yet standard in the particle physics community. Examples where trained networks have been published in ONNX format for future reuse are the DNNLikelihood [123], a package for cross-disciplinary training of discriminator networks [124], and the ATLAS search for R-parity-violating supersymmetry [125, 126], the latter also being available in the ATLAS SimpleAnalysis framework. However, detailed documentation for instance of the input variables is missing. Further development is strongly encouraged for, e.g., the purpose of analysis preservation [127, 128], and in general for the implementation of the Findable, Accessible, Interoperable, and Reusable (FAIR) principles for scientific data management [129] of ML models. An example for a dataset with special emphasis on these aspects can be found in Ref. [130]. Making the newly developed simulation and analysis tools, along with the required data, accessible to other scientists and future users forms an essential element of open and thriving science.

# 6 Outlook

As a field combining vast datasets with excellent, first-principle simulations, particle physics is benefiting tremendously from developments in data science and machine learning. While new AI-inspired methods will not magically solve all challenges in LHC simulations and analysis, they are providing a crucial and transformative contribution to our numerical toolbox. Moreover, given the quality of the LHC datasets, simulations, and simulation-based analysis methods, we expect particle physics to eventually contribute to broader machine learning research.

Event generation, or the simulation of signals for the LHC detectors from QFT Lagrangians, is the main link between experimental and theoretical particle physics. It has stringent requirements when it comes to first principles vs modeling, control, precision, speed, and flexibility. In this review we have shown that even within the physics-motivated modular structure of standard event generators, there is no aspect that cannot be improved through modern machine

learning. This includes phase space sampling, scattering amplitudes, loop integrals, parton showers, parton densities, and fragmentation. Some of these ML-applications have a long history and are accepted as standard approaches, other ML-based improvements of physics modules are currently under rapid development and are finding their way into standard generators. All of them will be key to address the needs for example of the HL-LHC.

In addition to ML-enhanced event generators, an interesting application of generative neural networks are ML-generators at parton level and fast ML-detector simulations. They provide an excellent testing ground for phase space generators, precision networks, and inverted simulations. This includes conceptual developments in the field of generative networks, driven by LHC-specific requirements of controlling precision-generative networks as numerical tools and providing a full range of uncertainties. They allow us to define, produce, and encode datasets for phenomenological studies and serve as a compression for data entering experimental analyses.

The main conceptual advantage of ML-event generation is that simulations with generative networks are symmetric: given a fundamental physics model we can predict the probability distributions of LHC events over phase space, or we can predict the probability distributions of model parameters given observed LHC events. Different ML-approaches to simulation-based inference include classification-based methods, conditional generative networks as a direct inversion, or indirect ways of learning likelihood ratios. In combination, they will allow us to systematically use unfolding or inverted simulations at the HL-LHC, from particle identification and detector unfolding all the way to an event-wise matrix element method analysis.

Finally, there are many simulation-related questions in fundamental physics, where AI-methods allow us to make significant progress. Examples going beyond immediate applications to event generation include symbolic regression [131], sample and data compression [58, 132], detection of symmetries [133–136], and many other fascinating new ideas and concepts.

# Acknowledgments

Anja Butter, Gudrun Heinrich, and Tilman Plehn are supported by the Deutsche Forschungsgemeinschaft under grant 396021762 - TRR 257 Particle Physics Phenomenology after the Higgs Discovery. Lukas Heinrich is supported by the Excellence Cluster ORIGINS, which is funded by the Deutsche Forschungsgemeinschaft under Germany's Excellence Strategy - EXC-2094-390783311. Alexander Held is supported by the U.S. National Science Foundation (NSF) Cooperative Agreement OAC-1836650. Stefan Höche and Joshua Isaacson are supported by the Fermi National Accelerator Laboratory (Fermilab), a U.S. Department of Energy, Office of Science, HEP User Facility. Fermilab is managed by Fermi Research Alliance, LLC (FRA), acting under Contract No. DE–AC02–07CH11359. Jessica N. Howard is supported by the U.S. National Science Foundation under the grant DGE-1839285. Michael Kagan is supported by the US Department of Energy (DOE) under grant DE-AC02-76SF00515. Gregor Kasieczka and Felix Kling are supported by the Deutsche Forschungsgemeinschaft under Germany's Excellence Strategy – EXC 2121 Quantum Universe – 390833306. Sabine Kraml acknowledges support by the IN2P3 master project Théeorie – BSMGA and the joint ANR-FWF project PRCI SLDNP grant no. ANR-21-CE31-0023. Claudius Krause and David Shih are supported by DOE grant DOE-SC0010008. Rahool Kumar Barman and Dorival Gonçalves thank the U.S. Department of Energy for the financial support under grant number DE-SC 0016013. Benjamin Nachman is supported by the U.S. Department of Energy, Office of Science under contract DE-AC02-05CH11231. Tilman Plehn is supported by the Deutsche Forschungsgemeinschaft under Germany's Excellence Strategy EXC 2181/1 - 390900948 (the Heidelberg STRUCTURES Excel-

lence Cluster). Steffen Schumann acknowledges support from the German Federal Ministry of Education and Research (BMBF, grant 05H21MGCAB) and the German Research Foundation (DFG, project number 456104544). Rob Verheyen is supported by the European Research Council (ERC) under the European Union's Horizon 2020 research and innovation programme (grant agreement No. 788223, PanScales). Ramon Winterhalder is supported by FRS-FNRS (Belgian National Scientific Research Fund) IISN projects 4.4503.16.

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
