# Peer review of "Machine Learning and LHC Event Generation"

_SciPost Physics, doi:SciPost Phys. 14, 079 (2023)_

## Round 1 · Referee Report · Anonymous (Referee 1) · 2022-10-12

Report

The authors review recent progress in the application of artificial intelligence/machine learning to event generation at the LHC.

As far as I can tell, this submission is identical to the contribution of the same authors to the Snowmass process.

Since the authors submitted this review to SciPost Physics, and since it is a review rather than original work and does not meet any of the 4 'Expectations' outlined in the SciPost Physics criteria, I do not think this should be published here. This is only my opinion and I defer the final decision on this point to the editor in charge. The review certainly meets the 'General acceptance criteria'.

If the editor decides it is publishable, I would like the authors to address or implement the following: 1. remove the 'Executive summary ' section since it does not add value to a technical review. 2. The assertion in the last sentence of the first paragraph in the introduction seems too strong. Namely, the one that reads "... will help provide the simulations needed for the LHC...". Perhaps something softer like "... has the potential to provide..." is more fitting since, as far as I know, there are no production ready generative models at the moment (which are the subject of ref. [6] cited at the end of that sentence). 3. The toolbox of traditional tools for event generation for the LHC is very rich and diverse, yet the authors cite this literature very narrowly in refs. [1-5] . What is the the rationale behind this choice?

  • validity: -
  • significance: -
  • originality: -
  • clarity: -
  • formatting: -
  • grammar: -

Author:  Tilman Plehn  on 2022-12-28  [id 3191]

(in reply to Report 1 on 2022-10-12)
Category:
answer to question

We would like to thank the referee and accommodated all his/her requests. For the non-ML aspects of event generators we added the corresponding Snowmass-inspired review as Ref.[6].

---

## Round 2 · List of Changes

We would like to thank the referee and accommodated all his/her requests. For the non-ML aspects of event generators we added the corresponding Snowmass-inspired review as Ref.[6].

---

## Editorial Decision

published